# Simultaneous Bilateral Frontal and Bilateral Cerebellar Transcranial Direct Current Stimulation in Treatment-Resistant Depression—Clinical Effects and Electrical Field Modelling of a Novel Electrodes Montage

**DOI:** 10.3390/biomedicines10071681

**Published:** 2022-07-12

**Authors:** Giordano D’Urso, Michelangelo Dini, Marta Bonato, Silvia Gallucci, Marta Parazzini, Natale Maiorana, Marco Bortolomasi, Alberto Priori, Roberta Ferrucci

**Affiliations:** 1Unit of Psychiatry, Department of Neurosciences, Reproductive and Odontostomatological Sciences, University of Naples Federico II, 80131 Naples, Italy; giordano.durso@unina.it; 2Aldo Ravelli Research Center, Department of Health Science, University of Milan, 20142 Milan, Italy; michelangelo.dini@unimi.it (M.D.); natale.maiorana@unimi.it (N.M.); alberto.priori@unimi.it (A.P.); 3Institute of Electronics, Computer and Telecommunication Engineering (IEIIT), National Research Council of Italy (CNR), 20133 Milan, Italy; marta.bonato@ieiit.cnr.it (M.B.); silvia.gallucci@ieiit.cnr.it (S.G.); marta.parazzini@ieiit.cnr.it (M.P.); 4Department of Electronics, Information and Bioengineering (DEIB), Polytechnic University of Milan, 20133 Milan, Italy; 5Psychiatric Hospital “Villa Santa Chiara”, 37142 Verona, Italy; marcobortolomasi.vr@gmail.com; 6ASST-Santi Paolo e Carlo, Neurology Unit, 20142 Milan, Italy

**Keywords:** transcranial direct current stimulation (tDCS), treatment resistant depression (TRD), cerebellum, dorsolateral prefrontal cortex (DLPFC)

## Abstract

Depressive disorders are one of the leading causes of disability worldwide. Transcranial direct current stimulation (tDCS) is a safe, simple, non-invasive brain stimulation technique showing considerable effectiveness in improving depressive symptoms. Most studies to date have applied anodal tDCS to the left dorsolateral prefrontal cortex (DLPFC), in line with the hypothesis that depressed patients exhibit relative hypoactivity in the left DLPFC compared to the right. Considering the emerging role of the cerebellum in emotional processes, we aimed to study the effect of combining bilateral cerebellar tDCS with the commonly used bifrontal stimulation in patients with severe depression. This open-label pilot study entailed the simultaneous administration of bilateral cerebellar (anode over the left cerebellum, cathode over the right cerebellum) and bilateral frontal (anode over the left DLPFC, cathode over the right DLPFC) tDCS to patients (*N* = 12) with treatment-resistant depression. The 21-item Hamilton Depression Rating Scale (HDRS) and Beck’s Depression Inventory-II (BDI-II) were selected as outcome measures. Electric fields distribution originating from this novel electrode montage was obtained by a computational method applied to a realistic human head model. We observed a 30% reduction of both clinician-rated and self-reported severity of depressive symptoms after only five days (10 sessions) of treatment. Younger age was associated with greater clinical improvement. Adverse events were similar to those of the conventional electrodes montage. The modelling studies demonstrated that the electric fields generated by each pair of electrodes are primarily distributed in the cortical areas under the electrodes. In conclusion, the cerebellum could represent a promising adjunctive target for tDCS interventions in patients with TRD, particularly for younger patients.

## 1. Introduction

Depression is a significant clinical and public health concern. Depressive disorders have been estimated to be the third leading cause of disability worldwide and are responsible for more “years lost” to disability than any other condition [1]. This is partly a consequence of the large number of people suffering from depression (around 350 million, according to the World Health Organization), but another key cause is the fact that depressive disorders tend to last many years, as classic therapeutic interventions (pharmacological therapy and/or psychotherapy) are often unable to determine a complete remission [2,3]. In addition to lasting disability, depression is also linked to greater mortality [4].

Researchers have been focusing on developing novel treatments for depression. While there is evidence that modern electroconvulsive therapy (ECT) can be effective in patients with TRD [5,6], its significant side effects, the need for general anaesthesia, and ideological prejudices against this therapy [7] have been pushing researchers towards less intensive procedures. Transcranial direct current stimulation (tDCS) is a safe, simple, non-invasive stimulation technique that does not require general anaesthesia, and consists of the application of weak direct electrical current to the brain through sponge electrodes placed on the scalp. This results in a shift of membrane resting potentials toward depolarisation (anodal stimulation) or hyperpolarisation (cathodal stimulation) [8]. High-quality evidence indicates that anodal tDCS applied over the left dorsolateral prefrontal cortex (DLPFC) is effective in producing clinically significant improvements in depressive symptoms [9,10,11,12], supporting the hypothesis that depressed patients exhibit a hypoactivity in the left DLPFC and relative hyperactivity in the right DLPFC [13]. In addition to the DLPFC, however, other brain regions and cortico–subcortical circuits are likely to play a significant mediating role in depressive symptoms [14,15,16,17].

The cerebellum is strongly connected to the cerebral cortex [18], and not only plays a major role in motor control [19] but also in emotional processes, given its extensive connections with important components of the emotional behaviour network, such as the amygdala, the hippocampus, and the prefrontal cortex (PFC) [20,21,22,23]. In fact, the cerebellum has been proposed as a pivotal component of the circuitry of several neuropsychiatric disorders [24,25,26], and particularly depression, a condition with consistent evidence of structural and functional cerebellar alterations [27,28,29]. Despite this, there is a scarcity of studies exploring the possible effectiveness of cerebellar stimulation in reducing the symptoms of depression. In a sham-controlled study on seventy-nine healthy adults, Newstead and colleagues reported that the stimulation of the left DLPFC combined with that of the right cerebellum significantly increased the mood of participants independently from the electrode’s polarity [30]. Studies involving euthymic bipolar patients indicated that anodal tDCS of the left DLPFC combined with cathodal tDCS of the right cerebellum can improve sleep [31], neuropsychological performance [32], and neurophysiological parameters [33] in this category of patients. The only study that targeted the cerebellum to reduce the symptoms of depression reported a non-significant clinical improvement in seven treatment-resistant depressed patients after twenty sessions of one left-prefrontal anodal tDCS combined with one cerebellar cathodal tDCS over the midline [34]. In designing the present study, we considered both the consolidated evidence of the efficacy of bilateral frontal tDCS and the involvement of the cerebellum in emotional processes. Therefore, we decided to combine bilateral cerebellar and bilateral frontal tDCS in a sample of patients with treatment-resistant depression (TRD). To the best of our knowledge, this is the first study to assess the safety and the antidepressant effect of a tDCS protocol involving two couples of electrodes simultaneously stimulating two different brain regions, each of which bilaterally.

## 2. Materials and Methods

### 2.1. Study Sample

Twelve patients (aged 32–75, 10 females) with TRD were recruited from the inpatient unit of the “Villa Santa Chiara” Psychiatry Clinic in Verona, Italy. TRD was defined as the failure to achieve symptomatic remission (total HDRS score ≤ 7) after treatment with either two different antidepressant drugs or one drug and cognitive-behavioural therapy [35].

The study was conducted in accordance with the Declaration of Helsinki (1964). Written informed consent was obtained from all participants before inclusion in the study, and the protocol was approved by the local ethics committee.

### 2.2. Stimulation Protocol

Direct current was generated by two electrical stimulators connected to two pairs of saline-soaked surface sponge electrodes (5 × 5 cm, 0.3 cm thick). The two electrodes connected to the first stimulator were applied to the prefrontal region (anode over the left DLPFC, cathode over the right DLPFC) and the two electrodes connected to the second stimulator were applied to the cerebellum (anode over the left cerebellum, cathode over the right cerebellum). Stimulation was administered simultaneously to the two electrode pairs with an intensity of 2 mA (current density: 0.08 mA/cm^2^) for 20 min, twice per day, at least four hours apart, for five consecutive days.

### 2.3. Electric Field Distribution Estimation

Electric fields (**E**) distributions due to the combined bilateral frontal plus bilateral cerebellar tDCS were obtained by a computational method applied to a realistic human head model. Simulations were obtained using the low-frequency solver of the simulation platform Sim4life (by ZMT Zurich Med Tech AG, Zurich, Switzerland, www.zurichmedtech.com (accessed on 1 May 2022)). The electric potential (*φ*) was obtained by solving the Laplace equation (∇⋅(σ∇φ)=0) where *σ* is the electrical conductivity of the human tissues. The **E** distributions were obtained by means of the following relations: E=−∇φ.

We used a multimodal imaging-based anatomical model, named MIDA [36] (see Figure 1), of the head and neck of one healthy 29-year-old female volunteer, segmented and reconstructed from three different MRI modalities at a 500 μm isotropic resolution. The head model distinguishes a total of 153 different regions, involving a high number of brain structures. We assigned the dielectric properties of the head tissues according to data collected in the IT’IS low-frequency tissue properties database [37]. The electrodes were modelled as squared pads of copper (*σ* =5.9 × 10^7^ S/m) with a squared sponge (*σ* = 1.4 S/m) which rested directly under the electrode. The potential difference between each pair of the electrodes was adjusted to inject a total current of 2 mA. During the simulation, the human model and the electrodes were inserted in a surrounding bounding box filled with air. All the boundaries were treated as insulated with exception of the truncation section to which boundary conditions of continuity of the current were assigned. Uniform rectilinear grid (mesh step = 0.5 mm) was used to discretize the computational domain.

### 2.4. Clinical Assessment

Patients underwent a psychometric evaluation during the week before starting (T0) and during the week after the completion (T1) of the tDCS treatment.

As a primary outcome measure, we used the 21-item Hamilton Depression Rating Scale (HDRS) [38], which provides a quantitative measure of clinician-rated depression severity. The HDRS can be divided into 6 factors: “Anxiety/somatisation”, “Weight loss”, “Cognitive impairment”, “Diurnal variation of mood”, “Psychomotor retardation”, and “Sleep symptoms” [39]. Depression severity can be categorised based on HDRS score (no depression = 0–7; mild depression = 8–16; moderate depression = 17–23; severe depression ≥ 24) [40].

As a secondary outcome measure, we administered the Beck Depression Inventory-II (BDI-II), to assess the effect of tDCS on self-reported severity of depressive symptoms. The BDI-II is a self-report questionnaire, comprising 21 items that can be divided into “somatic-affective” items, (e.g., “Tiredness”) and “cognitive” items, (e.g., “Past failure”) [41]. Severity of symptoms can be classified as minimal (scores < 13), mild (scores 13–19), moderate (scores 20–28), or severe (scores > 29) [42].

For the assessment of adverse events, we used the tDCS questionnaire surveying for adverse effects (tDCS-QSAE) [43].

### 2.5. Statistical Analyses

Given the small sample size, we opted for a nonparametric approach. We performed Wilcoxon signed-ranks tests to evaluate differences in HDRS and BDI-II scores between T0 and T1. Effect sizes for Wilcoxon signed-ranks test were calculated as $r=\;Z/\left(\sqrt{N\;\mathrm{of}\;\mathrm{observations}}\right)\;$ [44]. We also calculated percentage changes at T1 for both measures (score% change = (T1 score − T0 score)/T0 score), with negative values indicating lower scores at T1. We analysed the effect of age on psychological scores using Spearman’s correlation coefficient (*r_s_*). We did not apply a correction for multiple comparisons given the small sample size and the exploratory nature of the study [45].

## 3. Results

### 3.1. Electric Field

Figure 1 shows two views of the four electrodes positioning on MIDA (Figure 1A), whereas the normalised **E** amplitude distributions on the cortex and cerebellum are shown in Figure 1B. The modelling studies demonstrated that electric field distribution generated by each pair of electrodes primarily targets the anatomical region to which they were applied, similarly to when they are applied separately.

### 3.2. HDRS

Based on the HDRS score at T0, 7/12 (58.3%) patients had severe depression, 4/12 (33.3%) had moderate depression, and one patient had mild depression. The mean HDRS score at T0 was 25.33 ± 6.43 (mean ± SD).

We observed a significant improvement after tDCS (T0 vs. T1 [mean ± SD] = 25.33 ± 6.43 vs. 18.50 ± 8.90, Z = 2.83, *p* = 0.002, *r* = 0.578). The mean improvement at T1 was 29.68 ± 23.54%. After tDCS, 4/12 (33.3%) patients still had severe depression, 2/12 (16.7%) had moderate depression, 5/12 (41.7%) had mild depression, and one patient achieved symptomatic remission.

We also analysed the effect of tDCS on each of the six factors of the HDRS. We observed a significant effect of tDCS on items assessing psychomotor retardation (T0 vs. T1 = 8.67 ± 2.53 vs. 6.33 ± 2.99, Z = 2.66, *p* = 0.027, *r* = 0.543) and anxiety/somatisation (T0 vs. T1 = 6.83 ± 2.82 vs. 5.00 ± 3.52, Z = 2.32, *p* = 0.027, *r* = 0.474), but not on other factors of the HDRS (See Table 1).

Age did not correlate with HDRS score at T0 (*p* = 0.724), but we found a strong correlation between age and HDRS score% change after tDCS (*r_s_* = 0.872, *p* = 0.001) (Figure 2).

Data are displayed as mean and standard deviation (SD) of scores before (T0) and after (T1) stimulation. HDRS = Hamilton Depression Rating Scale; BDI-II = Beck’s Depression Inventory-II. *p*-values reflect the statistical significance of differences between T0 and T1 scores.

### 3.3. BDI-II

At T0, all but one patient reported severe depressive symptoms; the mean BDI-II score at T0 was 40.75 ± 10.66.

BDI-II scores improved significantly after tDCS (T0 vs. T1 = 40.75 ± 10.66 vs. 28.08 ± 13.16, Z = 3.06, *p*< 0.001, *r* = 0.626). The mean improvement at T1 was 32.71 ± 20%. At T1, symptoms were reported as severe by 33.3% of patients, moderate by 41.7%, and mild by 16.7%; one patient reported minimal symptoms (Figure 3).

Differences between T0 and T1 scores were significant for both the “cognitive” factor (T0 vs. T1 = 18.08 ± 4.76 vs. 11.25 ± 6.58, Z = 2.85, *p* = 0.002, *r* = 0.582) and the “somatic-affective” factor (T0 vs. T1 = 22.67 ± 6.64 vs. 16.83 ± 7.67, Z = 2.91, *p* = 0.002, *r* = 0.594).

At T0, we did not observe significant correlations between age and BDI score, (total BDI score: *p* = 0.400; “cognitive” factor: *p* = 0.637; “somatic-affective” factor: *p* = 0.495). Age did not correlate with total BDI-II score change after tDCS (*p* = 0.184), nor with changes in the two BDI-II sub-scores (cognitive: *p* = 0.900; somatic/affective: *p* = 0.178).

### 3.4. tDCS-QSAE

The reported adverse effects were skin redness (8/12 patients, 66.7%), tingling (8/12 patients, 66.7%), itching (6/12 patients, 50%), burning sensation (3/12 patient, 25%), sleepiness (2/12 patients, 16.7%), discomfort (2/12 patient, 16.7%), and headache (1/12 patient, 8.3%). None of them was rated as “severe” nor made it necessary to interrupt the treatment or modify the protocol.

## 4. Discussion

Our study aimed to assess the safety and the antidepressant effect of combined bilateral frontal plus bilateral cerebellar tDCS in patients with TRD. The first important finding is that the number and severity of adverse events were similar to that of conventional antidepressant tDCS with bifrontal electrodes montage [46]. As to the efficacy of this novel intervention, we observed a significant improvement in both clinician-rated and self-reported severity of symptoms after only five days of stimulation (T1) (Figure 3). This suggests that bilateral tDCS of the cerebellum might be safe and effective as an add-on treatment to the classical bilateral frontal tDCS montage to induce a more rapid and complete symptom reduction in patients with TRD.

While only one patient achieved full remission, the overall improvement after stimulation was noticeable, as the percentage of patients with moderate to severe depression decreased from 91.7% to 50% after only one week of stimulation, 3/12 (25%) achieved clinical response (as defined by an improvement ≥ 50%), and 6/12 (50%) improved by ≥30%.

In terms of mean scores improvement, our results are similar to those of previous studies using the bifrontal montage with the anode over the left DLPFC and the cathode over the right DLPFC or orbitofrontal area [9,10,11,12]. Due to the design of our study, we cannot disentangle the relative contribution of DLPFC and cerebellar tDCS, which is an issue that needs to be properly addressed by future research. Nevertheless, our data suggest that improvements can be observed in patients with TRD by combining prefrontal and cerebellar stimulation. This is particularly relevant, since relatively recent evidence-based guidelines had not recommended bilateral DLPFC tDCS for depression due to insufficient evidence, while a level B recommendation was given only for non-TRD patients and for anodal left DLPFC with right orbitofrontal cathode [47]. A possible explanation of the non-superior efficacy of our combined four-electrode montage, compared to the conventional bifrontal one, could be that our sample included only TRD patients, for which the other montages showed lower or no effect.

The observed mean improvement of about 30% in both HDRS and BDI-II scores was greater than the improvement described by Ho and colleagues (2014) [34], who reported a 15.9% decrease in depressive symptom severity after twice as long treatment (twenty daily sessions over four weeks) of fronto-cerebellar tDCS, an outcome similar to that of the sham stimulation in RCTs [48].

The difference between the results of Ho’s study and ours might be due to the different number, size and position of the cerebellar electrodes. First, while we applied an anode on the left side and a cathode on the right side of the cerebellum, Ho and colleagues applied a single electrode, i.e., a cathode, on the cerebellar midline. Therefore, while we probably induced a differential modification of excitability over the cortex of the two cerebellar lobes (increase on the left and decrease on the right), they induced an inhibition on both sides. This could have contributed to the different outcomes, considering also that our computational model showed that the electric fields induced by our montage are specifically located under the cerebellar electrodes [49]. Moreover, the cerebellum exerts an inhibitory control (cerebello–brain inhibition—CBI) over the prefrontal areas [50], mainly through crossed fibres. Considering the hypothesis of an asymmetrical activity of the two DLPFC in depression (left hypoactive, right hyperactive) [13], we assume that in our four-electrode montage, the anodal stimulation of the left cerebellum could have increased the cerebellar inhibition over the right DLPFC, while the cathodal tDCS over the right cerebellum could have reduced the cerebellar inhibition over the left DLPFC, thus boosting the effects of the bilateral DLPFC electrodes in restoring the symmetry of bilateral frontal activity. This hypothesis is in line with a recent study suggesting that the anodal stimulation of DLPFC combined with the cathodal stimulation of the right cerebellum could restore the asymmetry between crossed and homolateral CBI in patients with autism spectrum disorder (ASD) [51]. Based on these considerations, we hypothesise that the lack of antidepressant effect of Ho’s montage was due to the simultaneous stimulation of both the right and left cerebellum by means of one single midline cathode and that the consequent inhibition of bilateral CBI could have led to a null effect on the bilateral prefrontal areas. In addition, the greater dimension of the cerebellar cathode used in their study (10x5 cm), compared to ours (5x5 cm), could have led to a different effect on the underlying cerebellar cortex, due to a lower current density.

Interestingly, we also observed that age was strongly correlated with HDRS score improvement after tDCS. In particular, we observed the greatest improvement in those under the age of 50 (Figure 1). This suggests that neural plasticity mechanisms could be a key factor in determining the efficacy of combined bilateral frontal plus bilateral cerebellar tDCS in depressed patients. Indeed, various mechanisms of synaptic plasticity are present in the cerebellum [52], which probably accounts for the long-lasting modulation of cerebral activity [53]. Nevertheless, one should keep in mind that the presence of severe depressive symptoms in older patients is correlated to longer disease duration and, most often, to treatment refractoriness. Moreover, since the baseline depression severity of our sample was high, we cannot rule out that we would not obtain a similar outcome with patients showing a lesser severe depression, considering that some studies indicate a differential effect of tDCS on patients with different baseline depression severity, with the best outcomes for the most severe ones [54].

Further research is also necessary to investigate the possible advantages of using this novel four-electrode montage for specific subsets of symptoms of depression. For example, it is possible that stimulating the cerebellum together with the frontal areas might yield an additional beneficial effect on sleep disorders, often associated with depression. In fact, sleep disorders are not a good predictor of response to bifrontal tDCS in depression [55], and, on the other side, sleep modulation is one of the nonmotor functions of the cerebellum [56]. In line with this, recent studies have shown that prefrontal–cerebellar tDCS, with the cathode over the right cerebellum and the anode over the left DLPFC, improved sleep quality in euthymic bipolar [31] and ASD [51] patients. Of note, in the latter clinical population, such a beneficial effect on sleep quality was not described when the tDCS protocol did not include the stimulation of the cerebellum [57,58]. Another possible avenue of investigation could focus on the influence of tDCS on the neuropsychological performance of patients with depression. In fact, while the beneficial effect of anodal left prefrontal tDCS on working memory is widely recognised (independently from the position of the cathode and the population under investigation) [59], there are studies showing a detrimental effect of bifrontal montage on some other aspects of neurocognition, i.e., implicit learning [60]. Of interest, the additional stimulation of the cerebellum was shown to improve mental processes coordination, a function specifically mediated by the cerebro–cerebellar circuit, central to task-switching [32]. We hypothesise that a deeper knowledge of the differential neuropsychological effects of the various tDCS montages might lead to the identification of the most beneficial treatment for the cognitive impairment of patients with depression, or even to a personalised intervention according to the specific cognitive profile of each patient.

Finally, our pilot study has some important limitations: we lacked a control condition, our sample was small and not balanced according to demographic variables, and we did not assess the long-term effects of stimulation. Therefore, further case–control trials with larger sample sizes, adequate stratification of demographic and clinical variables, and longer follow-ups are needed to validate our promising preliminary results, thus providing new targets and protocols for the non-pharmacological treatment of depression.

## 5. Conclusions

In conclusion, the results of this study provide preliminary clinical evidence that stimulating both the cerebellum and the DLPFC by means of a double bilateral montage might represent a promising approach for the treatment of TRD, with younger patients probably being the best candidates. Moreover, this novel tDCS protocol could have a special indication for more severely depressed patients with sleep disorders and/or cognitive impairment.

## Figures and Tables

**Figure 1 biomedicines-10-01681-f001:**
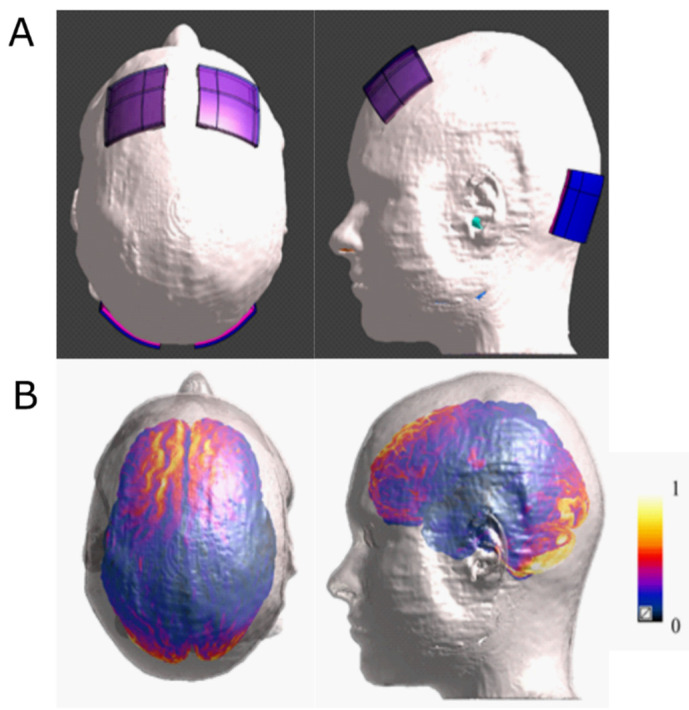
Computational model of electrical field distribution: (**A**) top and lateral views of the electrode positioning on the MIDA model; (**B**) top and lateral views of the normalised electrical field distributions below the stimulating electrode on the cortex and cerebellum. The distributions are normalised with respect to the maximum of the electric field amplitude.

**Figure 2 biomedicines-10-01681-f002:**
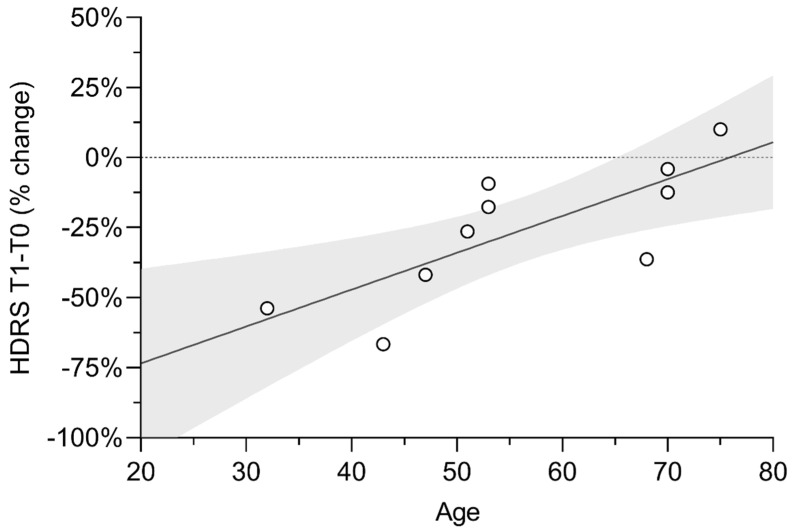
Correlation between age and HDRS improvement after tDCS scatterplot with linear fit showing a positive correlation between age and Hamilton Depression Rating Scale (HDRS) score% change after cerebellar transcranial direct current stimulation (tDCS). Y–axis values reflect the percentage change of HDRS score after stimulation, compared to baseline scores (negative values indicate lower scores following tDCS, compared to baseline). Dots represent individual cases; the light-grey area represents the 95% CI of the linear fit.

**Figure 3 biomedicines-10-01681-f003:**
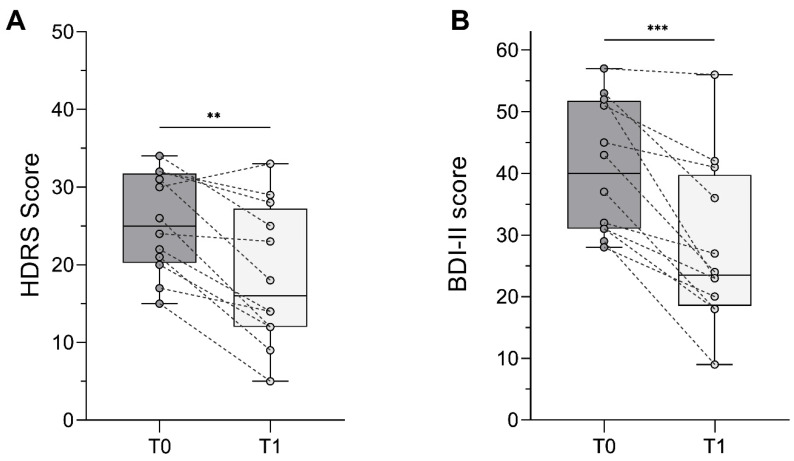
Effect of combined bilateral frontal plus bilateral cerebellar tDCS on depressive symptoms. Boxplots show decrease in Hamilton Depression Rating Scale (HDRS) score (**A**) and Beck Depression Inventory-II (BDI-II) score (**B**) after tDCS. Dots represent individual cases; dashed lines connect T0 (pre-stimulation) and T1 (post-stimulation) scores of the same subject. Asterisks denote statistically significant differences (** = *p*< 0.01, *** = *p*< 0.001).

**Table 1 biomedicines-10-01681-t001:** Differences between psychological scores at T0 and T1.

	T0 Score		T1 Score		Z	*p*-Value
	Mean	SD	Mean	SD		
**HDRS**						
Total Score	25.33	6.43	18.50	8.90	2.83	**0.002**
Anxiety/Somatisation	6.83	2.82	5.00	3.52	2.32	**0.027**
Weight Loss	0.58	0.99	0.42	0.90	0.82	0.750
Cognitive Impairment	4.17	2.79	3.33	2.50	1.27	0.254
Diurnal Variation of Mood	1.67	1.50	1.42	1.50	0.55	0.813
Psychomotor Retardation	8.67	2.53	6.33	2.99	2.66	**0.006**
Sleep Symptoms	3.42	2.47	2.00	2.41	1.97	0.055
**BDI-II Total Score**						
Total Score	40.75	10.66	28.08	13.16	3.06	**<0.001**
Cognitive Factor	18.08	4.76	11.25	6.58	2.85	**0.002**
Somatic-Affective factor	22.67	6.64	16.83	7.67	2.91	**0.002**

## Data Availability

Data is contained within the article.

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
