# Peer review of "Simultaneous Bilateral Frontal and Bilateral Cerebellar Transcranial Direct Current Stimulation in Treatment-Resistant Depression—Clinical Effects and Electrical Field Modelling of a Novel Electrodes Montage"

_biomedicines, 2022, doi:10.3390/biomedicines10071681_

Round 1
Reviewer 1 Report
As described on lines 302-303 in the text, lacking a control study is critical limitation.
In this case, another sham control group (electrode only, no current) is at least necessary.
Author Response
We thank the reviewer for giving us the opportunity to clarify this important point. Due to the severity of baseline depression and the history of suicide attempts of many patients in our sample, we could not include a sham control group (Thase, 1999). However, the severity of baseline depression, the magnitude of improvement, and the resistance to previous standard treatment, make a placebo effect unlikely. Nevertheless, future studies should enroll less severely depressed patients and include a sham control group to confirm our preliminary results. We have now included this information in the discussion.
Reviewer 2 Report
In this paper, the effect of multi (10) session tDCS, delivered to bilateral prefrontal cortex and celebellum, was found to improve depressive symptoms in patients with treatment-resistant depression. The number of subjects was 12, and young participants showed relatively better response. This is a open-label one-arm pilot study.
The positive results would add to the literature for the potential ability of tDCS to alleviate an array of symptoms in psychiatric illnesses.
This reviewer would like to raise only some formatting issues, as follows.
1. One of the lengthy paragraphs (linew 257 -288) could be made more concise.
2. The final 3 paragraphs (lines 305 -344) should be re-organized or re-ordered, so that the sentences for conclusions of this paper will be gathered.
Author Response
- One of the lengthy paragraphs (linew 257 -288) could be made more concise.
Reply: We thank the reviewer for this suggestion. We revised those paragraphs accordingly, by making that part more straightforward and concise.
- The final 3 paragraphs (lines 305 -344) should be re-organized or re-ordered, so that the sentences for conclusions of this paper will be gathered.
Reply: We thank the reviewer for bringing this to our attention. We have revised the final paragraphs accordingly.
Round 2
Reviewer 1 Report
OK, I'm looking forward to meeting the authors' future works.